# Influence of Different Types of Retinal Cameras on the Performance of Deep Learning Algorithms in Diabetic Retinopathy Screening

**DOI:** 10.3390/life12101610

**Published:** 2022-10-15

**Authors:** Ramyaa Srinivasan, Janani Surya, Paisan Ruamviboonsuk, Peranut Chotcomwongse, Rajiv Raman

**Affiliations:** 1Shri Bhagwan Mahavir Vitreoretinal Services, Sankara Nethralaya (Main Campus), No 41 (Old 18), College Road, Chennai 600006, Tamil Nadu, India; 2Department of Ophthalmology, Rajavithi Hospital, College of Medicine, Rangsit University, Bangkok 12000, Thailand

**Keywords:** diabetic retinopathy, artificial intelligence, retinal camera, retinal images

## Abstract

**Background:** The aim of this study was to assess the performance of regional graders and artificial intelligence algorithms across retinal cameras with different specifications in classifying an image as gradable and ungradable. **Methods:** Study subjects were included from a community-based nationwide diabetic retinopathy screening program in Thailand. Various non-mydriatic fundus cameras were used for image acquisition, including Kowa Nonmyd, Kowa Nonmyd α-DⅢ, Kowa Nonmyd 7, Kowa Nonmyd WX, Kowa VX 10 α, Kowa VX 20 and Nidek AFC 210. All retinal photographs were graded by deep learning algorithms and human graders and compared with a standard reference. **Results:** Images were divided into two categories as gradable and ungradable images. Four thousand eight hundred fifty-two participants with 19,408 fundus images were included, of which 15,351 (79.09%) were gradable images and the remaining 4057 (20.90%) were ungradable images. **Conclusions:** The deep learning (DL) algorithm demonstrated better sensitivity, specificity and kappa than the human graders for all eight types of non-mydriatic fundus cameras. The deep learning system showed, more consistent diagnostic performance than the human graders across images of varying quality and camera types.

## 1. Introduction

Diabetic Mellitus is one of the fastest-growing metabolic disorders in the world. The International Diabetes Federation has estimated that the global prevalence of diabetes is expected to reach 693 million by 2045 [1]. A serious microvascular complication associated with diabetes is diabetic retinopathy (DR), a leading cause of blindness in adults. With a rising incidence of diabetes, there is an associated increase in microvascular complications like diabetic retinopathy (DR). In patients with diabetes, regular follow-up and timely management are essential to preventing visual impairment due to sight-threatening DR. Resnikoff et al. reported that 17% of the global population has access to less than 5% of the worldwide ophthalmologist population [2]. A higher national income was associated with a higher ophthalmologist density, ranging from 76.2 per million in high-income countries to 3.7 per million in low-income countries [2]. People in low-to-middle-income countries are at significant risk of developing DR as a result of resource constraints, i.e.,. In contrast, sufficient shortage of eye specialists and limited infrastructure for disease detection and management are unavailable. Retinal imaging is important in screening, monitoring and managing DR.

Artificial intelligence (AI), which can detect the presence of DR, has the potential to enhance and automate aspects of DR screenings, and recent efforts have examined their clinical applicability. In recent years several studies have validated the use of deep learning for DR screening and have demonstrated robust performance [3,4,5,6]. However, the development and validation of many of these algorithms have been done using retinal images from a single specification of a retinal camera. Thus the generalizability of the algorithm’s performance to other cameras may be questionable. To the best of our knowledge, the influence of different types of retinal cameras on the performance of deep learning algorithms has not been studied in the past. The algorithm’s performance is closely related to the fundus camera on which it has been trained and eventually deployed. This could be due to differences in colour reproduction, image clarity, different fields of view, and penetration of opacity by a given fundus camera. For this reason, the regulatory authorities tend to specify the camera to be used to get optimum performance of AI algorithms for DR screening [7,8].

Thailand has a national DR screening program that was set up by the Ministry of Public Health in 2013. The program enables non-physician healthcare personnel (mainly nurses) to manually conduct eye screenings in primary care clinics and community hospitals. The goal of the program is to aid in the early detection of diabetic eye disease and ensure timely referral of diabetes patients to retinal specialists for management. The program has different types of retinal cameras deployed for screening. The present study is a secondary analysis of the original dataset from the Thai national screening program that compared the performance of human graders with AI grading for DR [9].

The aim of this study was to assess the performance of regional graders and AI algorithm across retinal camera’s with different specifications in classifying an image as gradable Vs ungradable and classifying DR based on international Clinical Diabetic Retinopathy (ICDR) severity scale.

## 2. Materials and Methods

Study subjects were included from a community—based nation wide DR screening program organized by the Thai Ministry of Public Health. The screening was conducted in 13 regions that include hospitals and health care centers in Thailand. For the current study, patients with diabetes were randomly included from the national registry between the years 2015 and 2017. Patients were included if they had fundus images of both the eyes captured using retinal cameras from both years, 2015 and 2017. A variety of non-mydriatic fundus cameras were used for image acquisition including Kowa Nonmyd, Kowa Nonmyd α-DⅢ, Kowa Nonmyd 7, Kowa Nonmyd WX, Kowa VX 10 α, Kowa VX 20 and Nidek AFC 210. The resolution of the fundus cameras ranges from 2 megapixel to 18 megapixels. Specifications of the non-mydriatic fundus cameras are summarized in Table 1.

Images were retrieved from the digital archives of the retinal cameras. Images were excluded if the patient had any other associated retinal disease except DR. All retinal photographs were graded by two groups of retina specialists or ophthalmologists for the standard reference. The retinal specialists that served as the standard reference were from Thailand, India and the United States. There were two retinal specialists per group. Each group graded the images independently. Images were divided into two categories as gradable or ungradable image. Images were labelled as ungradable if the both eyes were ungradable, or if either eye was ungradable. The same set of images were separately graded by the deep learning algorithm and by human graders that included general ophthalmologists, trained ophthalmic nurses or technicians. The development, evaluation and validation of deep learning algorithm was described in detail elsewhere [10]. In summary, if the retinal image didn’t show the key regions with good enough quality for a confident grading and the macula is not visible or only partially visible (less than one disc diameter from the centre of the fovea) either because it is not in the field of view or it is occluded by artefacts, dark shadow etc. and what is seen is no enough to rule out DME were labelled as ungradable. (Figure 1) A tutorial session was conducted for all the graders before the commencement of grading to ensure standardization of grading methodology. The study design and research methodology has been described in detail in a previous publication elsewhere [9]. The study was approved by the Ethical Review Committee for research in human subjects by the Thai Ministry of Public Health, Thailand and by those the Ethical Committees of hospitals and health care centers from which retinal images of patients with diabetes were used. A written informed consent was obtained from the subjects per the tenets of the Declaration of Helsinki.

## 3. Statistical Analysis

Statistical analysis was performed using statistical software package (SPSS for Windows, version 21.0; IBM Corp., Armonk, NY, USA). The data were tested for normality using the Kolmogorov-Smirnov (K-S) test. The results were expressed as number and percentage for categorical data and continuous data were expressed as mean with standard deviation. Independent samples t-tests were used to check for the existence of a significant difference in normally distributed data, and Mann–Whitney U tests were used for non-normally distributed data. The performances of different cameras in assessing gradable and ungradable images were measured by the AUC of the receiver operating characteristic curve generated by plotting sensitivity (the true-positive rate) versus 1-specificity (the false-negative rate). The AUCs were compared using binary classification methods. Kappa (κ) statistics were used to assess the inter-observer agreement (i.e., the agreement of the human graders with a standard reference and deep learning algorithm with a standard reference). A well-known scale was used for interpretation of results (0–0.20, slight agreement; 0.21–0.40, fair agreement; 0.41–0.60, moderate agreement; 0.61–0.80, substantial agreement; 0.81–1.00, almost perfect agreement). For all the analysis a two-sided P value of less than 0.05 was considered statistically significant.

## 4. Results

In our study, 4852 participants with 19,408 fundus images were included. Demographic details are available for only 4588 subjects with a mean age of 59.19 ± 11.52 years; 1524 (33.2%) were men and 3063 (66.8%) were women. The demographic characteristics of the participants are summarized in Table 2. Table 3 shows the patient distribution of DR severity as graded by human graders, ophthalmologist and the algorithm.

Of the total 19,408 fundus images included in the study, 15,351 (79.09%) were gradable images and the remaining 4057 (20.90%) were ungradable images. The number of gradable and ungradable images from each retinal camera is shown in Table 4.

The diagnostic performance of the human graders and deep learning algorithm vs. the standard reference in the different fundus cameras are shown in Table 5. In both gradable and ungradable images, the deep learning algorithm demonstrated better sensitivity, specificity and kappa than the human graders for all eight types of non-mydriatic fundus cameras. Figure 2 and Figure 3 show the receiver operating characteristic (ROC) curve of the model for gradable and ungradable images by human graders (A) and deep learning algorithm (B) vs. standard reference for non-mydriatic fundus cameras. For gradable images, the deep learning algorithm shows high accuracy (ROC > 90) than human graders (ROC: 40–90) whereas in ungradable images both deep learning algorithm (ROC: 40–95) and human graders (ROC: 25–90) demonstratee similar results. The best AUC’s in detecting sight threatening DR (Moderate + grade of DR) in the gradable images was obtained by Kowa Nonmyd 7 fundus camera with DL as compared to standard reference, AUC = 0.95 (0.92–0.97) *p* < 0.0001. Likewise, for detection of ungradable image the performance of Nidek AFC 210 was the best; AUC: 0.98 (0.97–0.99) *p* < 0.0001.

## 5. Discussion

Early detection and timely treatment are essential in order to avoid vision loss due to DR. The recent advent of AI-based algorithms on fundus imaging has made it easier to increase patient access to DR screening and timely referral of individuals with sight-threatening DR to ophthalmologists. In this study we assessed the influence of non-mydriatic fundus cameras of different specifications on the diagnostic performance of the deep learning algorithm to diagnose referable DR in comparison with the regional graders. The diagnostic performance and agreement was compared between the deep learning algorithm and the standard reference in the assessment of grading fundus images. We assessed the performance of human graders and deep learning algorithm in both gradable and ungradable images.

We report here that the deep learning algorithm shows a high sensitivity, specificity and kappa (both gradable and ungradable images) than human graders relative to the standard reference for the detection of DR with images taken from fundus cameras of different specifications. This is similar to the results of Gulshan et al., where the same deep learning algorithm used images from a variety of cameras, including Centervue DRS, Optovue iCam, Canon CR1/DGi/CR2, and Topcon NW. EYEPACS-1 shows 97.5% sensitivity and 93.4% specificity whereas Messidor-2 shows 96.1% sensitivity and 93.9% specificity [10].

Ting et al. validated a deep learning system from the Singapore national DR Screening Program to detect referable DR, vision threatening DR, and related eye diseases (referable possible glaucoma and referable AMD) [5]. Their analysis used a range of retinal cameras with a resolution of 5–7 megapixels. The sensitivity and specificity for referable DR was 90.5% (95% CI 87.3–93.0%) and 91.6% (95% CI 91.0–92.2%) respectively, and for STDR the sensitivity and specificity were 100% (95% CI 94.1–100.0%) and 91.1% (95% CI 90.7–91.4%) respectively. The deep learning system showed consistent diagnostic performance across images of varying quality and different camera types. The dataset included poorer quality images, including ungradable ones, which resulted in a somewhat lower performance of the deep learning system (AUC, 0.936). However, the diagnostic accuracy of the deep learning system based on the different camera types was not done.

One of the main reasons for varying sensitivity in both gradable and ungradable images might be due to the visibility of lesions from different types of fundus cameras. In addition, the graders experience level, knowledge, differences in the room setup, lighting conditions and the visibility of lesions may vary in the images taken from different types of fundus cameras as a result of the different resolutions.

To the best of our knowledge, this is the first study that has looked at the influence of non-mydriatic retinal cameras with different specifications on the performance of DL algorithm and human graders in both gradable and ungradable images. This study also has a few limitations. The sample size is lower in the category of ungradable images and patient demographics are not the same across the different cameras. Furthermore, the analysis does not include images taken from mydriatic fundus cameras and smartphone-based fundus cameras as the study was not designed to explore this possibility. It would also be good to compare newer network models or other deep learning network models which might show a better performance across the camera’s. The ideal data set to compare the performance of AI algorithm across different camera would be the same patient images in different camera’s. However, in real world setting, the results from the present study seem to suggest a better performance of AI algorithm as compared to regional grades across different cameras. Future prospective studies are required to address these limitations.

Our study demonstrated that the deep learning algorithm performs better than the human graders, irrespective of images taken from fundus cameras of different specifications with both gradable and ungradable images.

## Figures and Tables

**Figure 1 life-12-01610-f001:**
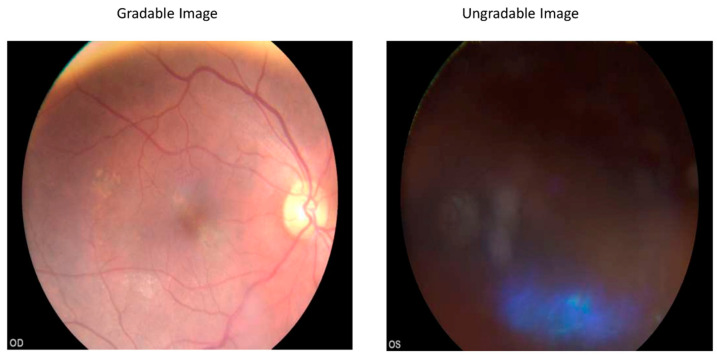
Sample images of gradable and ungradable images.

**Figure 2 life-12-01610-f002:**
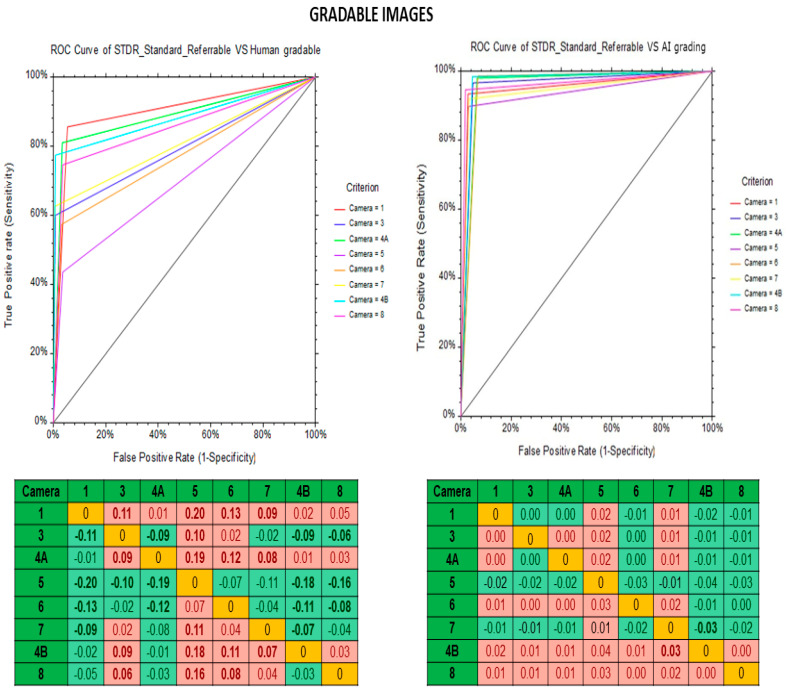
Receiver operating characteristic (ROC) curve of model for gradable images in different types of non-mydriatic fundus cameras. A—human graders vs. standard reference and B—deep learning algorithm vs. standard reference.

**Figure 3 life-12-01610-f003:**
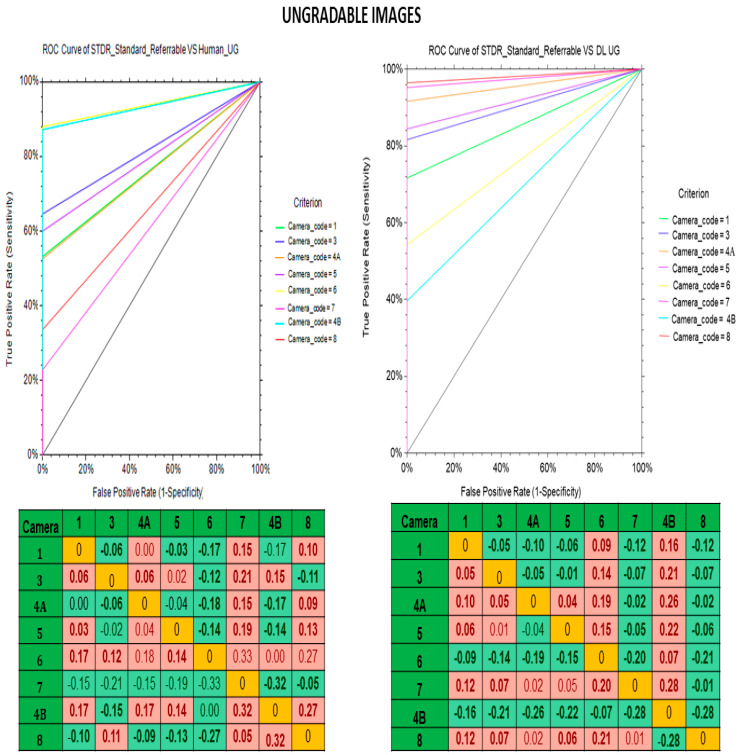
Receiver operating characteristic (ROC) curve of model for ungradable images in different types of non-mydriatic fundus cameras. A—human graders vs. standard reference and B—deep learning algorithm vs. standard reference.

**Table 1 life-12-01610-t001:** Specifications of the non-mydriatic fundus cameras.

Specifications	Kowa Nonmyd α-DⅢ	Kowa Nonmyd 7	Kowa Nonmyd WX	Kowa VX 10 α	Kowa VX 20	Nidek AFC 210
Field angle	45°/30°	45°/20°	Normal: 45° Stereo: 34° (20° × 27°) SP:45°	-	-	45°
Internal fixation target	Normal: 3 positions (central, disc, macula); Mosaic: 9 positions	4 fixed dots switching type	Central, Disc, Macula, Mosaic: 8 positions	4 fixed dots right or left eye switching (Non-mydriatic mode)	Central, Disc, Macula, Peripheral	LED (70 points)
Minimum pupil size	3.5 mm	4.0 mm (small pupil mode: 3.7 mm)	Normal mode: 4.0 mm, SP mode: 3.5 mm, Stereo mode: 4.0 mm	Non-mydriatic mode- 4.00 mm; Mydriatic mode- 5.5 mm; Small pupil- 4.00 mm	Non-mydriatic mode- 4.00 mm; Mydriatic mode- 5.5 mm; Small pupil- 4.00/3.5 mm	4.0 mm (Small pupil: 3.7 mm)
Focusing	Split luminous bars coincidence	Split luminous bars coincidence	Split luminous bars coincidence	Point matching method (ON/OFF switch)	Split luminous bars coincidence	Infrared split bright target coincidence
Compensation range of examined eye	Without compensation: −15D to +13D; With—compensation: −32D to −12D; With + compensation: +10D to +40D	Without compensation: −15D to +13D; With—compensation: −33D to −11D; With + compensation: +10D to +40D	Without compensation: −12D to +13D; With—compensation: −32D to −10D; With + compensation: +10D to +35D	Without compensation −12D to +13D; With—compensation: −10D to −32D; With + compensation: +10D to +35D	Without compensation −12D to +13D; With—compensation: −10D to −32D; With + compensation: +10D to +35D	Total: −33D to +35D; With minus dioptric lens: −33D to −7D; With no dioptric lens: −12D to +15D; With plus dioptric lens: +11D to +35D

**Table 2 life-12-01610-t002:** Demographic details of the study subjects.

Parameter	All	Gradable Images	Ungradable Images	*p*-Value
No of subjects	4588	3816	772	**-**
Age, mean ± SD	59.19 ± 11.52	57.81 ± 11.17	66.01 ± 10.76	**<0.001**
Gender, *n* (%)				
Male	1524 (33.2)	1282 (33.6)	242 (31.3)	**<0.001**
Female	3063 (66.8)	2533 (66.4)	530 (68.7)	**<0.001**
HbA1c, (%)	5.56 ± 3.70	5.51 ± 3.37	5.80 ± 3.55	**<0.001**
Serum FBS, (mg/dL)	117.61 ± 77.17	117.21 ± 77.15	119.61 ± 77.27	0.431
Serum LDL, (mg/dL)	79.93 ± 57.68	80.39 ± 58.43	77.62 ± 53.77	**0.001**
Visual acuity (logMAR), median(IQR)				
Right eye	0.17 (0.39)	0.17 (0.30)	0.30 (0.37)	**<0.001**
Left eye	0.17 (0)	0.17 (0)	0.30 (0)	**<0.001**
Both eyes	0.17 (0.30)	0.17 (0.35)	0.30 (0.34)	**<0.001**

FBS = Fasting Blood Sugar, LDL = Low-Density Lipoprotein, HbA1c: Glycosylated hemoglobin.

**Table 3 life-12-01610-t003:** Distribution of DR severity by human graders, Ophthalmologists and algorithm.

	Ophthalmologist (N%)	Human Graders (N%)	DL (N%)
No DR	12,648 (82.39)	13,225 (86.15)	12,376 (80.62)
Mild NPDR	1172 (7.63)	811 (5.28)	647 (4.21)
Moderate NPDR	1239 (8.07)	1081 (7.04)	1456 (9.48)
Severe NPDR	94 (0.61)	78 (0.51)	402 (2.62)
PDR	198 (1.29)	156 (1.02)	170 (1.11)

DR: Diabetic Retinopathy, NPDR: Non proliferative DR, PDR: Proliferative DR.

**Table 4 life-12-01610-t004:** Gradable and ungradable images from different non-mydriatic retinal cameras.

Camera Name	Camera Code	Resolution (Megapixel)	Gradable Images N = 15,351 *n* (%)	Ungradable Images N = 4057 *n* (%)
Kowa Nonmyd	1	2	1383 (9.0)	1039 (25.6)
Kowa Nonmyd α-DⅢ	3	8	3241 (21.1)	775 (19.1)
Kowa Nonmyd 7	4A	10	1895 (12.3)	344 (8.5)
Kowa Nonmyd WX	5	12	733 (4.8)	45 (1.1)
Kowa VX 10 α	6	12.3	728 (4.7)	118 (2.9)
Kowa VX 20	7	15	3133 (20.4)	409 (10.1)
Kowa Nonmyd 7	4B	16	1249 (8.1)	493 (12.2)
Nidek AFC 210	8	18	2989 (19.5)	834 (20.6)

**Table 5 life-12-01610-t005:** Comparison of sensitivity, specificity and kappa for gradable and ungradable images by human graders and deep learning algorithm vs. standard reference in different fundus cameras.

Fundus Camera	Gradable Images						Ungradable Images					
	Human Graders vs. Reference Standard			Deep Learning Algorithm vs. Standard Reference			Human Graders vs. Reference Standard			Deep Learning Algorithm vs. Standard Reference		
	Sensitivity	Specificity	Kappa	Sensitivity	Specificity	Kappa	Sensitivity	Specificity	Kappa	Sensitivity	Specificity	Kappa
	(95% CI)	(95% CI)		(95% CI)	(95% CI)		(95% CI)	(95% CI)		(95% CI)	(95% CI)	
Kowa Nonmyd	85.6	94.6	0.72	93.5	97.2	0.85	53.2	98.7	0.37	71.5	99.7	0.97
	(79.0–90.8)	(93.2–95.9)		(88.3–96.8)	(96.1–98.1)		(47.8–58.6)	(97.8–1.0)		(66.4–76.2)	(99.8–1.0)	
Kowa Nonmyd α-DⅢ	59.9	99.1	0.68	96.6	95.1	0.72	64.5	98.9	0.75	81.6	99.7	0.88
	(53.3–66.3)	(98.7–99.4)		(93.3–98.5)	(94.3–95.9)		(61.0–67.9)	(97.8–1.0)		(78.6–84.2)	(99.1–1.0)	
Kowa Nonmyd 7	81	96.7	0.8	97.9	93.7	0.86	52.5	99.5	0.66	91.5	99	0.81
	(76.9–84.6)	(95.6–97.5)		(96.1–99.0)	(92.3- 94.8)		(43.2–61.8)	(99.1–1.0)		(85.0–95.9)	(98.9–1.0)	
Kowa Nonmyd WX	43.6	96.4	0.39	89.7	97	0.72	60	99.7	0.74	84.4	98.7	0.91
	(27.8–60.4)	(94.7–97.7)		(75.8–97.1)	(95.4–98.1)		(44.3–74.3)	(99.5–1.0)		(70.5–93.5)	(98.4–1.0)	
Kowa VX 10 α	57.4	96.7	0.61	98.7	93.3	0.84	88	99.3	0.66	54.3	99.7	0.95
	(49.1–65.5)	(94.9–98.0)		(95.2–99.8)	(90.9–95.2)		(84.5–91.0)	(99.1–1.0)		(49.3–59.2)	(99.1–1.0)	
Kowa VX 20	62.6	99.5	0.72	91.9	96.9	0.75	22.7	99.9	0.93	95.1	99.1	0.67
	(55.5–69.4)	(99.2–99.7)		(87.2–95.3)	(96.2–97.5)		(19.1–26.7)	(99.7–1.0)		(92.8–96.9)	(97.4–1.0)	
Kowa Nonmyd 7	74.6	96.5	0.68	94.6	98.2	0.88	30.7	98.2	0.3	96.4	98.9	0.97
	(65.4–82.4)	(95.3–97.5)		(88.5–98.0)	(97.3–98.9)		(65.4–36.6)	(98.0–1.0)		(95.1–97.5)	(98.6–1.0)	
Nidek AFC 210	77.4	99.2	0.82	98.4	95.4	0.8	84.8	99.4	0.91	39.6	99.7	0.51
	(72.3–82.0)	(98.8–99.5)		(96.2–99.5)	(94.6–96.2)		(72.3–89.5)	(99.2–1.0)		(36.2–43.0)	(99.2–1.0)	

For all the four comparisons, *p* < 0.001.

## Data Availability

The data that support the findings of this study are available from the corresponding author, [author initials], upon reasonable request.

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
