# Peer review of "Influence of Different Types of Retinal Cameras on the Performance of Deep Learning Algorithms in Diabetic Retinopathy Screening"

_life, 2022, doi:10.3390/life12101610_

Round 1

Reviewer 1 Report

The main purpose of this paper is to evaluate the performance of deep learning methods on classifying fundus images of different specifications. By using the Inception-v3 convolutional neural network in reference 8, classification results close to or even better than human ratings are obtained on the dataset. The study has positive significance for promoting the application of deep learning in the screening of diabetic retinopathy. The idea of this paper is good, aiming to find a general automatic classification method. The structure of this article is complete, but some content can be improved, and it is recommended to be revised.

1. It is necessary to explain the differences in the fundus images obtained from different cameras. This involves the underlying logic of this article. For example, a certain deep learning method performs better on the images from one machine, but it does not perform well on images from another machine.

2. Reference 8 was published in 2016. Can it be verified and compared on newer network models or other deep learning network models? Deep learning is developing very fast, it will maybe get better results using other network models?

3. There is a lack of introduction to scalable pictures and non-gradable pictures. This is the main object of the verification task, and giving typical pictures and corresponding textual explanations can help readers better understand what you have done.

4.The graphs and tables in figure 1 and figure 2 are not clear enough, and I suggeste to redraw them.

Author Response

Review Report (Reviewer 1)

Comments and Suggestions for Authors The main purpose of this paper is to evaluate the performance of deep learning methods on classifying fundus images of different specifications. By using the Inception-v3 convolutional neural network in reference 8, classification results close to or even better than human ratings are obtained on the dataset. The study has positive significance for promoting the application of deep learning in the screening of diabetic retinopathy. The idea of this paper is good, aiming to find a general automatic classification method. The structure of this article is complete, but some content can be improved, and it is recommended to be revised.

  1. It is necessary to explain the differences in the fundus images obtained from different cameras. This involves the underlying logic of this article. For example, a certain deep learning method performs better on the images from one machine, but it does not perform well on images from another machine.

Response 1: Thanks for the suggestion. We have added a para to describe the logic of this article in the introduction.

  1. Reference 8 was published in 2016. Can it be verified and compared on newer network models or other deep learning network models? Deep learning is developing very fast, it will maybe get better results using other network models?

Response 2: We agree with the reviewer, that newer network models can probably give better results. However, they have not yet tried on different cameras. We have added this to the discussion.

  1. There is a lack of introduction to scalable pictures and nongradable pictures. This is the main object of the verification task, and giving typical pictures and corresponding textual explanations can help readers better understand what you have done.

Response 3:  Thanks for the suggestion. In the methods section, we have added details of classifying the image as gradable or non-gradable.

  1. The graphs and tables in figure 1 and figure 2 are not clear enough, and I suggested to redraw them.

Response 4: High-resolution image has been added.

Please see the attachment for the revised manuscript

Reviewer 2 Report

Dear Authors,

The objective of the algorithm is only to assess whether the images can be graded or not, or is it able of grade them (Diabetic Retinopathy Severity Scale, DRSS)?

Those are my sugestions and corrections:

1) Line 84:

Both the years-->remove "the"

2) Tables:

Review all tables, as content of tables doesn't match citation in text.

Line 89: Table 1 shows demographics, not fundus camera specifications.

Line 132: Table 2 shows gradable and ungradble images, not demographis.

Line 137: Table 3 shows camera specifications, not gradable and ungradable images.

Please add abbreviations meaning in all tables as a footnote to the table (FBS=fasting blood sugars,LDL= low-density lipoprotein .....)

3) Line 92:

Only images with another retinal disseases were excluded? There were no image quality criteria ( lens opacity, focus, clarity)?

4)Line 98:

Please provide further information about what "gradable and ungradable" means. As an eyecare professional I supose that they have graded the different Diabetic Retinopathy Severity Scale (DRSS) scores or any similar scale, but this paper is also interesting for other health care professionals, so they should also understand first, what thay have graded and wich scale has been used and second, what "gradable and ungradable" means.

5)Line 99:

Why images were ungradable? Because there was not DR?

6)Table 3:

There are several data provided on Table 3, that really does not add anything of interest to the paper (Weight,Dimensions, interface, chin rest adjustment...). The analysis of the characterisitics is not the main goal of this study, so all this space could be used to provide important statistical info.

7)Results suggestions:

It will enrich the paper to add basic statistical info, such as gradin gresults.

How many Mild NPDR, Moderated NPDR, Severe NPDR and PDR patients [n=(%)] were graded by each group (Standard of reference ophthalmologist, Human graders and DL algorithm)?

8) Figures 1&2:

Images are blurry, it's really difficult to read the caption.

9) Discussion:

The title of the paper is "the influence of different types of cameras", so is there any camera that has obtained the best ROC values ​​for both humans and the algorithm? Because ROC varies depending on the camera.

Author Response

Review Report (Reviewer 2)

  1. Comments and Suggestions for Authors Dear Authors, The objective of the algorithm is only to assess whether the images can be graded or not, or is it able of grade them (Diabetic Retinopathy Severity Scale, DRSS?

Response : The aim of this study was to assess the performance of regional graders and AI algorithm across retinal camera’s with different specifications in classifying an image as gradable Vs ungradable and classifying DR and DME based on the international Clinical Diabetic Retinopathy (ICDR) severity scale. This has been added in the introduction

Those are my suggestions and corrections:

  1. Line 84: Both the years-->remove "the"

        Response 1: Thanks, the correction has been done in the revised manuscript.

  1. Tables: Review all tables, as the content of the tables doesn't match the citation in the text.

Response 2 : Sorry, we have corrected this in the revised manuscript.

Line 89: Table 1 shows demographics, not fundus camera specifications.

Response : Sorry, we have corrected this in the revised manuscript.

Line 132: Table 2 shows gradable and ungradable images, not demographics

Response: Sorry, we have corrected this in the revised manuscript.

Line 137: Table 3 shows camera specifications, not gradable and ungradable images.

Response: Sorry, we have corrected this in the revised manuscript.

Please add abbreviations meaning in all tables as a footnote to the table (FBS=fasting blood sugars,LDL= low-density lipoprotein .....)

Response : Abbreviations have been added in the tables.

  1. Line 92: Only images with another retinal disseases were excluded? There were no image quality criteria ( lens opacity, focus, clarity)?

Response 3: Yes, only other retinal diseases were excluded. Those with poor quality were also included in the study.

  1. Line 98: Please provide further information about what "gradable and ungradable" means. As an eyecare professional I suppose that they have graded the different Diabetic Retinopathy Severity Scale (DRSS) scores or any similar scale, but this paper is also interesting for other health care professionals, so they should also understand first, what thay have graded and wich scale has been used and second, what "gradable and ungradable" means.

Response 4: The definitions of the ungradable image is included in the revised manuscript. Details of grading DR have also been added.

  1. Line 99: Why images were ungradable? Because there was not DR?

Response 5: No, the details of the ungradable have been included.

  1. Table 3: There are several data provided on Table 3, that really does not add anything of interest to the paper (Weight, Dimensions, interface, chin rest adjustment...). The analysis of the characteristics is not the main goal of this study, so all this space could be used to provide important statistical info.

Response 6: Thanks for the suggestion. We have included only the parameters which will have an influence on the image quality in the revised table.

  1. Results suggestions: It will enrich the paper to add basic statistical info, such as gradin results.  How many Mild NPDR, Moderated NPDR, Severe NPDR and PDR patients [n=(%)] were graded by each group (Standard of reference ophthalmologist, Human graders and DL algorithm)?

Response 7: Thanks for the suggestion. We have added this information as a new table (Table 2).

  1. Figures 1&2: Images are blurry, it's really difficult to read the caption.

Response 8: New high resolution images have been added in the revised manuscript.

  1. Discussion: The title of the paper is "the influence of different types of cameras", so is there any camera that has obtained the best ROC values for both humans and the algorithm? Because ROC varies depending on the camera.

Response 9: This information has been added in the revised manuscript in the results section.

"Please see the attachment for the revised manuscript"

Round 2

Reviewer 2 Report

Thanks for all the corrections. Now everything looks more clear. The paper has improve so much.